# Beyond Single Stationary Policies: Meta-Task Players as Naturally Superior Collaborators

**Haoming Wang**[*,1]**, Zhaoming Tian**[*,1]**, Yunpeng Song**[1]**, Xiangliang Zhang**[2]**, Zhongmin Cai**[†,1]

[1]MOE KLINNS Lab, Xi'an Jiaotong University, Xi'an, Shaanxi, China
[2]Department of Computer Science and Engineering, University of Notre Dame, Notre Dame, IN, USA
{wanghm,tzm9802}@stu.xjtu.edu.cn, yunpengs@xjtu.edu.cn, xzhang33@nd.edu,
zmcai@sei.xjtu.edu.cn

## Abstract

In human-AI collaborative tasks, the distribution of human behavior, influenced by mental models, is non-stationary, manifesting in various levels of initiative and different collaborative strategies. A significant challenge in human-AI collaboration is determining how to collaborate effectively with humans exhibiting non-stationary dynamics. Current collaborative agents involve initially running self-play (SP) multiple times to build a policy pool, followed by training the final adaptive policy against this pool. These agents themselves are a single policy network, which is **insufficient for handling non-stationary human dynamics**. We discern that despite the inherent diversity in human behaviors, the **underlying meta-tasks within specific collaborative contexts tend to be strikingly similar**. Accordingly, we propose **C**ollaborative **B**ayesian **P**olicy **R**euse (**CBPR**[1]), a novel Bayesian-based framework that **adaptively selects optimal collaborative policies matching the current meta-task from multiple policy networks** instead of just selecting actions relying on a single policy network. We provide theoretical guarantees for CBPR's rapid convergence to the optimal policy once human partners alter their policies. This framework shifts from directly modeling human behavior to identifying various meta-tasks that support human decision-making and training meta-task playing (MTP) agents tailored to enhance collaboration. Our method undergoes rigorous testing in a well-recognized collaborative cooking simulator, *Overcooked*. Both empirical results and user studies demonstrate CBPR's superior competitiveness compared to existing baselines.

## 1   Introduction

An ongoing challenge in artificial intelligence (AI) involves training agents capable of effective collaboration with humans Klien et al. [2004], Bard et al. [2020], Dafoe et al. [2020]. Unlike typical AI-only multi-agent collaboration, human-AI collaborative scenarios such as two-player cooking games, autonomous driving, and managing power grid stability incorporates a non-stationary component, humans Jagerman et al. [2019], Chandak et al. [2020], Chandak [2022]. As humans may vary in their level of initiative, alter their collaboration strategies, or sometimes even do not collaborate at all. This variability suggests that for cooperative agents, the probability distribution $P(A|s_t)$ of a human action $A$ given an environmental state $s_t$ changes over time, reflecting different

---

[*]Equal contribution.
[†]Corresponding author.
[1]We make our code publicly available https://github.com/AlexWanghaoming/CBPR

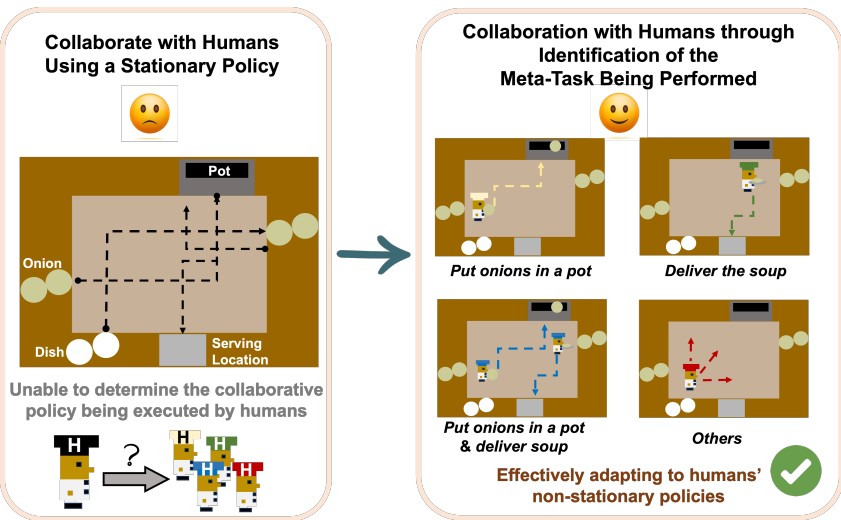

Figure 1: *Left*: The drawbacks of current collaborative agents, which train a stationary policy to manage the non-stationary dynamics of human collaborators but fail to determine the specific collaborative policies executed by humans. *Right*: Our approach focuses on identifying the meta-tasks underlying human decision-making and trains collaborators to match these meta-tasks in a one-to-one manner. This strategy enables effective ad-hoc collaboration with non-stationary humans.

mental states. Such non-stationarity poses a significant challenge in training collaborative agents, as it requires strategies that can adapt to the unpredictable nature of human behavior, which departs from the stable action-outcome associations expected in scenarios dominated by AI.

Recent works mainly develop collaborative agents through two workflows: (1) explicitly model human behavior by using real human trajectories Carroll et al. [2019], and then train a collaborator by teaming up with human models. (2) train Self-Play (SP) agents to form a policy pool (a diverse set of AI agents assumed to encompass all potential human policies) and then train a collaborator pairing with policies in the policy pool Strouse et al. [2021], Yu et al. [2023], Zhao et al. [2023]. However, despite their ability to achieve commendable performance by amassing extensive human data collection or SP agent training, these collaborators share a common fundamental flaw: *they are essentially policy networks following a stationary distribution, thus making it difficult to cope with non-stationary human dynamics.*

In this work, we propose Collaborative Bayesian Policy Reuse (CBPR), which reuses multiple stationary policies tailored to meta-tasks within a specific collaborative scenario. CBPR builds upon Bayesian Policy Reuse (BPR) Rosman et al. [2016], Chen et al. [2022], extending its application to human-AI collaborative tasks with theoretical guarantees. CBPR avoids modeling the non-stationary dynamics of human collaborators, focusing instead on heuristically modeling available meta-tasks within defined collaborative contexts. For example, in the multi-player cooking game *Overcooked*, meta-tasks include {*place onions in pot*, *deliver soup*, *place onions in pot & deliver soup*, *others*} (Figure 1) are available. Noticing that for a complex human-AI collaborative task, all of the undefined meta-tasks are categorized as "others," we subsequently train stationary meta-task-playing (MTP) collaborators using reinforcement learning (RL) to precisely match meta-task models on a one-to-one basis. During collaboration, CBPR identifies the meta-task being performed by the human partner based on recent actions, subsequently adapting the optimal MTP collaborator for use.

We evaluate CBPR in a fully-observable two-player common-payoff collaborative cooking simulator based on the game Overcooked Carroll et al. [2019], which has recently been proposed as a coordination challenge for AI Carroll et al. [2019], McKee et al. [2022], Wang et al. [2020], Wu et al. [2021], Knott et al. [2021]. State-of-the-art performance of this game was achieved in Carroll et al. [2019], Strouse et al. [2021], Yu et al. [2023] via training stationary cooperation policy. Both simulated experiments and user studies show that the proposed CBPR agent can collaborate effectively with non-stationary agents and real humans. The novel contributions of this paper can be summarized as follows:

1. We introduce a human-AI collaboration framework, CBPR, which addresses the challenge of modeling non-stationary human dynamics. This framework identifies the meta-tasks performed by human partners and reuses the optimal collaborative policy.

2. Theoretically, based on the Non-Stationary Markov Decision Process (NS-MDP), we provide theorems on *Collaboration Convergence* and *Collaboration Optimality* to support CBPR's convergence to the optimal collaborative policy over time in human-AI collaboration.

3. Empirically, we demonstrate CBPR's capability to collaborate effectively with non-stationary agents who frequently switch strategies, agents with various collaboration skills, and real humans.

## 2 Related Work

### 2.1 Human-AI Collaboration

Training agents to collaborate with humans has been extensively studied. Recent research can be categorized into two groups based on whether human data is used during training. BCP Carroll et al. [2019] is trained by pairing with a supervised human model, while Boltzmann Policy Distribution (BPD) Laidlaw and Dragan [2022] updates its prior based on online human actions. These approaches require human data collection and are prone to distributional shifts. In contrast, another category focuses on achieving zero-shot coordination without extensive human data Hu et al. [2020]. These works (e.g., FCP Strouse et al. [2021], Hidden-Utility Self-Play (HSP) Yu et al. [2023], and Maximum Entropy Population-based Training (MEP) Zhao et al. [2023]) train Self-Play (SP) agents to form a policy pool—a diverse set of AI agents assumed to encompass all potential human policies—and then train a collaborator to pair with policies in this pool. However, these collaborative agents remain single *stationary* models despite their diverse training partners.

Our work represents a fundamental departure from previous studies by avoiding the need to model human behavior and instead focusing on constructing meta-tasks that underpin human decision-making. Furthermore, our CBPR framework does not restrict the construction of meta-tasks, which can be categorized into two streams: reliant on human data (e.g., behavior cloning) and independent of human data (e.g., rule-based methods).

### 2.2 Policy Reuse

Policy reuse is a kind of transfer learning method that can greatly speed up reinforcement learning for a new task by using policies for relevant tasks. Initial methods like PRQL Fernández and Veloso [2013] and OPS-TL Li and Zhang [2018], Li et al. [2018] integrated source policies with limitations in transfer efficiency. Subsequent approaches such as CAPS and CUP Zhang et al. [2022] improved policy selection and introduced more efficient algorithms without the need for extra training components.

Bayesian policy reuse (BPR) Rosman et al. [2016] represents a specialized stream within policy reuse. Utilizing a Bayesian optimization approach, BPR efficiently computes posteriors for novel tasks. Extensions like BPR+ Hernandez-Leal et al. [2016a,b] and Bayes-Pepper Hernandez-Leal and Kaisers [2017] adapt BPR to multiagent scenarios, aligning tasks with opponent strategies and policies with optimal responses to these strategies. However, most BPR methodologies Rosman et al. [2016], Hernandez-Leal et al. [2016a], Hernandez-Leal and Kaisers [2017], Zheng et al. [2018, 2021], Chen et al. [2022], Xie et al. [2022] primarily address multi-task problems or copy with competitive scenarios. Several studies, such as Zheng et al. [2018, 2021], investigated deep BPR+ in collaborative games.

However, these approaches primarily rely on policy inference to adjust to the changing strategies of opponents (or partners), which may not be optimal for human-AI collaboration given the wide spectrum of potential human policies. To our knowledge, our research is pioneering in applying and tailoring Bayesian policy reuse-based algorithms specifically for the human-AI collaboration challenge.

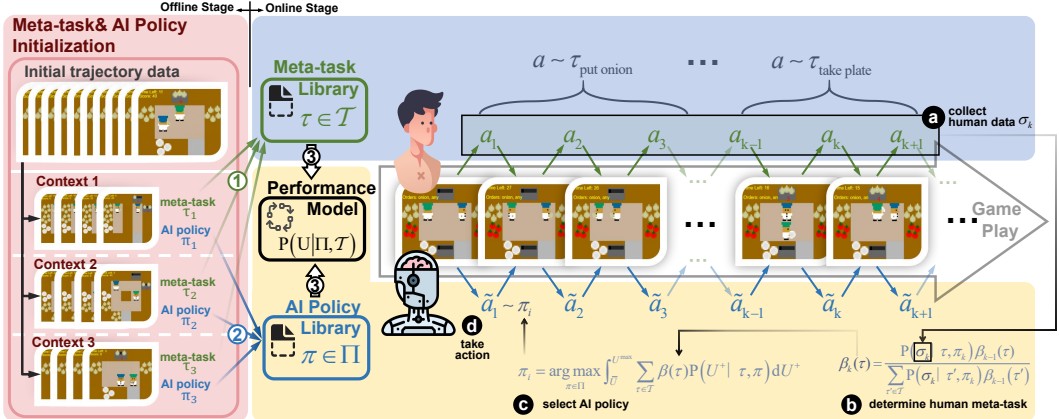

Figure 2: Overview of the CBPR Framework. This framework is divided into two main phases. *Left:* Offline Training Phase. This includes **(1)** constructing meta-task models using collected data and creating a meta-task library; **(2)** developing cooperative policies for each meta-task to compile an AI policy library; **(3)** establishing a performance model by evaluating each meta-task and AI policy pair. *Right:* Online Collaboration Phase. During a collaboration round, the process involves **(a)** gathering a list of historical and current human data; **(b)** determining the current meta-task undertaken by the human using Bayesian policy inference (refer to Equation 3-4); **(c)** selecting the most suitable AI policy for cooperation (as per Equation 5); and finally, **(d)** the AI collaborator executes actions according to the chosen policy.

## 3 Collaborative Bayesian Policy Reuse

### 3.1 Vanilla Bayesian Policy Reuse

Bayesian policy reuse is a general framework of transfer learning to cope with unknown tasks or frequently changing opponents. These classes of methods typically involve two phases: an offline learning phase and an online reusing phase. The workflow of a typical BPR can be summarized as follows: In the offline phase, it is presupposed that there exists a library of tasks $\mathcal{T}$ and a corresponding library of learned policies $\Pi$. Through conducting multiple simulations with varied policies across different tasks, a performance model $P(U \mid \mathcal{T}, \Pi)$ is derived, where $U = \Sigma_{i=0}^{k} r_i$ is cumulative utility. This model works as a mapping operator, associating each task and policy with a distribution of a predefined utility measure, such as reward.

During the online phase, BPR identifies the current task or opponent policy by maintaining a belief model $\beta(\cdot)$. This model is periodically updated based on observations, as defined by the observation model $P(\sigma \mid \tau, \pi)$, where $\sigma$ represents any signal aiding cooperation, such as reward or interaction trajectory. Significantly, this update adheres to Bayes' rule as follows:

$$\beta_k(\tau) = \frac{P(\sigma_k \mid \tau, \pi_k) \beta_{k-1}(\tau)}{\sum_{\tau' \in \mathcal{T}} P(\sigma_k \mid \tau', \pi_k) \beta_{k-1}(\tau')} \tag{1}$$

With this belief model, the BPR agent can select the optimal response policy by solving the following optimization problem:

$$\pi^\star = \mathrm{argmax}_{\pi \in \Pi} \int_{\bar{U}}^{U^{\mathrm{max}}} \sum_{\tau \in \mathcal{T}} \beta(\tau) P(U^+ \mid \tau, \pi) \, dU^+ \tag{2}$$

where $\bar{U} = \max_{\pi \in \Pi} \sum_{\tau \in \mathcal{T}} \beta(\tau) \mathbb{E}[U \mid \tau, \pi]$ represents the average performance of a single policy across all tasks. It's important to note that using $\bar{U}$ as the lower limit of the integral, this optimization problem essentially seeks the policy with the highest likelihood of achieving utility above the average.

### 3.2 CBPR Framework

**Offline stage** Initially, we train meta-task processing (MTP) agents $\pi \in \Pi$ using the Proximal Policy Optimization (PPO) algorithm by individually pairing them with meta-tasks within a specific

collaborative context, as exemplified by tasks such as *place onions in pot*, *deliver soup*, *place onions in pot & deliver soup*, and *others* in the *Overcooked* collaboration benchmark. Meta-task models $\tau \in \mathcal{T}$ are constructed through supervised learning, utilizing trajectories from either *rule-based agents enhanced with noise* or *real humans performing the tasks*. In this study, we employ the rule-based agents developed by Yu et al. [2023]. Subsequently, we construct the performance model $P(U \mid \mathcal{T}, \Pi)$ (i.e., observation model) by fitting a Gaussian distribution over the mean episodic return given a stochastic AI policy $\pi$ and a noisy rule-based agent $\tau$.

In previous BPR-based algorithms, the belief is designed for measuring the similarity between different tasks or opponents in transfer learning. These algorithms update belief using a observation model $P(\sigma \mid \tau, \pi)$ which only considers the game result but overlooks opponent's behavior. This leads to a poor collaborative performance when humans switch policy in a long-episode game. In this study, we used intra-episode belief $\xi^t(\tau)$ at timestep $t$ to measure the similarity between current meta-task $\tau$ and $\tau'$ in meta-task model library $\mathcal{T}$. The intra-episode belief was firstly proposed in Chen et al. [2022] and we extend it to the human-AI collaborative scenario.

**Online policy reuse** At the beginning of online policy reuse, the inter-episode belief $\beta_0(\tau)$ is initialized with a uniform distribution. For each episode, CBPR maintains a first-in-first-out (FIFO) human behavior queue $\mathcal{Q}$ of length $l$, which records the latest human behavior tuples $(s_i, a_i)$. The AI selects initial response MTP agents according to the inter-episode belief $\beta_0(\tau)$ (line 5 in Algorithm 1). CBPR collects human state-action pairs and updates the intra-episode belief $\xi_t(\tau)$:

$$\xi_t(\tau) = \frac{P(\mathcal{Q} \mid \tau)\xi_{t-1}(\tau)}{\sum_{\tau' \in \mathcal{T}} P(\mathcal{Q} \mid \tau') \xi_{t-1}(\tau')} \tag{3}$$

where $P(\mathcal{Q} \mid \tau) = \frac{\exp\left(\sum_{i=0}^{l} \log \tau(a_i|s_i)\right)}{\sum_{\tau' \in \mathcal{T}} \exp\left(\sum_{i=0}^{l} \log \tau'(a_i|s_i)\right)}$. Then the intra-episode belief and inter-episode belief are integrated:

$$\zeta_t(\tau) = \rho^t \beta_{k-1}(\tau) + \left(1 - \rho^t\right) \xi_t(\tau) \tag{4}$$

Where $\rho \in [0, 1]$ is a hyperparameter controlling the weight of the inter-episode and intra-episode beliefs. As the timestep $t$ increases in a game with a long episode, the integrated belief $\zeta_t(\tau)$ primarily depends on the intra-episode belief $\xi_t(\tau)$. The AI then uses the integrated belief $\zeta_t(\tau)$ to select a policy to cooperate with the human at each timestep.

$$\pi_t^\star = \arg \max_{\pi \in \Pi} \int_{\bar{U}}^{U^{\max}} \sum_{\tau \in \mathcal{T}} \zeta_t(\tau) P\left(U^+ \mid \tau, \pi\right) \mathrm{d}U^+ \tag{5}$$

At the end of each episode, CBPR collects the episodic return $u_k$ and updates the inter-episode belief $\beta_k(\tau)$. To adapt to non-stationary human dynamics, we store human-AI trajectories in a replay buffer $\mathcal{R}$ of the current MTP agent and update its policy. The detailed pseudo-code for the policy reuse of CBPR is presented in Algorithm 1.

### 3.3 Theory Analysis of CBPR

The selection of cooperative policies (line 11 in the Algorithm 1) is crucial to the performance of CBPR in collaborating with humans. In this section, we propose theorems on the convergence and optimality of CBPR to support our viewpoint: CBPR will converge to the optimal cooperative strategy during the human-AI interaction process. We formulate collaborative process between humans and AI as a Non-Stationary MDP (NS-MDP) Chandak et al. [2020]. In this process, the non-stationarity, resulting from the dynamic nature of human policy, can be mitigated by decomposing the entire non-stationary decision process into several stationary ones. Each stationary MDP corresponds to a specific meta-task executed by the human. Specifically, for a given NS-MDP $\{M_i\}_{i=1}^{\infty}$, the transition function integrates human actions as part of the environment itself, which can be denoted as $\mathcal{P}_i : \mathcal{S} \times \mathcal{A}_{AI} \times \mathcal{A}_{hu} \to \Delta(\mathcal{S})$. Within each stationary MDP $M_i$, the human policy $\pi_{hu,i} : \mathcal{S} \to \Delta(\mathcal{A})$ is assumed to be stationary, although it may exhibit variations across different stationary MDPs. Under this assumption, the CBPR agent could establish a convergent human-AI collaboration:

**THEOREM 1** (Collaboration Convergence of CBPR Agent). *Let $H_i := \{S_i^j, \pi_{hu,i}(S_i^j), R^j\}_{j=0}^{\infty}$ be a trajectory collected from a single stationary MDP $M_i$ within the overall NS-MDP $\{M_i\}_{i=1}^{\infty}$ under the human meta-task policy $\pi_{hu,i}$. Denote $\mathcal{D} := \{(i, H_i) : i \in [1, k]\}$ as a random variable representing a set of trajectories observed prior to the most recently completed stationary MDP $M_k$. Given $\mathcal{D}$, the*

**Algorithm 1** Online Policy Reuse of CBPR

**Input**: Meta-task model library $\mathcal{T}$, meta-task playing (MTP) agent library $\Pi$, performance model $P(U|\Pi,\mathcal{T})$, human behavior queue $\mathcal{Q} = \emptyset$, total timesteps $T$ in one episode

1: Initialize $\beta_0(\tau)$ with a uniform distribution
2: **for** episode k=1,2,3,...,K **do**
3:     Empty the queue $\mathcal{Q}$
4:     $\xi_0(\tau) \leftarrow \beta_{k-1}(\tau)$
5:     Select initial MTP agent $\pi$ to cooperate with human using Eq. 5
6:     **while** $t < T$ **do**
7:         Human chooses action $a_i$ and AI choose action according to $\pi(a \mid s)$
8:         Append the human behavior tuple $(s_t, a_t)$ to $\mathcal{Q}$
9:         Update belief $\xi_t(\tau)$ using Eq. 3
10:       Update integrated belief $\zeta_t(\tau)$ using Eq. 4
11:       Select a optimal MTP agent $\pi$ to cooperate with human in next timestep by using Eq. 5
12:       $\xi_t(\tau) \leftarrow \zeta_t(\tau)$
13:       $t \leftarrow t + 1$
14:     **end while**
15:     $\beta_k(\tau) \leftarrow \xi_T(\tau)$
16:     Update belief $\beta_k(\tau)$ using episodic return $u_k$ as observation signal following Eq. 1
17: **end for**

---

*response policy of CBPR agent could almost sure converge when interacting with a human partner, even when the human's policy is non-stationary.*

We provide all proofs and a detailed explanation in Appendix A. In addition to being able to converge in cooperation with non-stationary humans, the CBPR agent can also establish the optimal collaboration policy:

**THEOREM 2** (Collaboration Optimality of CBPR Agent). *Denoting* CBPR *for CBPR algorithm, let* $\rho(\pi, m) := \mathbb{E}[\int_{\tilde{U}}^{U^{\max}} \mathrm{P}\left(U^+ \mid \tau(m), \pi\right) \mathrm{d}U^+]$ *be the expected return of exploiting AI policy $\pi$ with human meta-task policy $\tau(m)$ in MDP $M_m$. Given a positive integer $k$ and a set of trajectories $\mathcal{D}$ observed prior to the MDP $M_k$, it follows that for any subsequent stationary MDP $M_{k+\delta}$, we have:*

$$\mathrm{Pr}\left(\rho\big(\mathrm{CBPR}(\mathcal{D}), k + \delta\big) \geq \rho(\pi_k^\star, k + \delta)\right) \to 1 \qquad (6)$$

*when $k \to \infty$, where $\pi_k^\star$ is the optimal response policy for human meta-task policy at MDP $M_k$.*

## 4 Experiments

In the context of *Overcooked*, we use rule-based policies developed in Yu et al. [2023] for each game layout (see Appendix C.1). These rule-based policies such as *place onions in pot*, *deliver soup* are used to train corresponding MTP agents in a one-to-one manner. In this section, we conduct extensive experiments to answer the following questions:

**Q1**: When interacting with non-stationary agents who switch their strategies, can CBPR outperform established baselines? Additionally, can CBPR adapt its collaborative strategies to better synchronize with partner behaviors?

**Q2**: When interacting with non-stationary agents of various collaboration skills, can CBPR surpass other baselines?

**Q3**: Can CBPR exceed the performance of other baselines in collaboration with real humans?

**Q4**: How do hyperparameters and number of predefined meta-tasks influence the collaborative performance (mean reward) of CBPR agents?

***Overcooked* environment** *Overcooked* is a popular two-player common-payoff game. It has become a typical environment for studying human-AI collaboration Carroll et al. [2019], Knott et al. [2021], Strouse et al. [2021], McKee et al. [2022], Yu et al. [2023]. In this game, players should place three onions or tomatoes in a pot and deliver as many cooked soups as possible within a time limit. Good coordination between two players is crucial for achieving a high score. We employed four layouts in

our experiments: *Cramped Room*, *Coordination Ring*, *Asymmetric Advantage* and *Soup Coordination* (Figure 8 in Appendix) in our experiments. Notably, in the *Asymm. Adv.* and *Soup Coord.*, the players do not interfere with each other, and their movements are unobstructed by their partners.

**Baselines** We compare CBPR against three well-established baselines: (1) the Behavioral Cloning Play (BCP) Carroll et al. [2019], a human model-based method designed for human-AI collaboration; (2) Fictitious Co-Play (FCP) Strouse et al. [2021], a two-stage approach trained with partners of varying skill levels; (3) Self-Play (SP) Silver et al. [2017], a common RL method trained by playing against itself. For a fair comparison, we employed PPO Schulman et al. [2017] as the underlying algorithm of CBPR and reimplemented all baselines using identical hyperparameters in our experiments. Further details about environment setting and agents are illustrated in Appendix C.

## 4.1 Cooperating with Rule-Based Agents Under Dynamic Policy Switching

To answer question **Q1**, we conduct a thorough investigation into the collaboration performance of CBPR when paired with non-stationary agents. These agents exhibited changes in their rule-based policies (Appendix Table 3), both inter-episodically and intra-episodically. We maintained a consistent random seed for policy switching during the evaluations to ensure fairness when comparing CBPR with baseline methods.

In our experiment, we evaluate the collaborative performance of agents at four different policy switching frequencies, as shown in Figure 3. The results show that CBPR consistently outperforms baseline methods in most cases. In particular, BCP, which was trained using a stationary human model, exhibited significantly poorer performance compared to CBPR. In addition, FCP and SP agents show greater fluctuations in episodic rewards, primarily due to their inability to effectively collaborate with all agents. In some instances, SP agents opted not cooperate, resulting in zero reward.

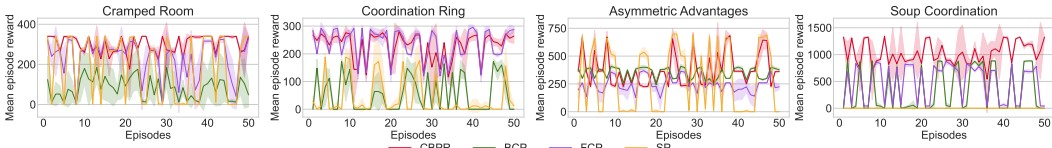

Figure 3: Comparative performance analysis against baselines when collaborators switch their rule-based policies *per episode*. All agents were evaluated over 50 continuous episodes. The shaded areas denote standard deviation calculated from five random seeds.

Our findings indicate that CBPR is particularly effective at collaborating with partners exhibiting varying degrees of non-stationarity. For a detailed overview of the results across the additional three policy switching frequencies (i.e., *per 2 episodes*, *per 200 timesteps*, and *per 100 timesteps*), please refer to the Appendix C.

## 4.2 Cooperation with Partners of Various Collaboration Skills

The cooperative capacity of non-stationary humans is typically suboptimal. A generalized agent must be capable of collaborating with partners possessing diverse collaboration skills.

During the initial training phase of FCP Strouse et al. [2021], a policy pool is created by preserving various agent "checkpoints" that represent different levels of expertise. To answer question **Q2**, we pair CBPR with agents with varying collaboration skills preserved during the first stage of the FCP training. We evaluate collaborative performance over 50 episodes on four layouts. The results show that CBPR consistently achieved higher mean episode rewards than FCP, particularly when collaborating with lower-skilled partners (Figure 4). It is noteworthy that BCP performs better in the *Asymm. Adv.* and *Soup Coord.* in which players' movements are not hindered by their partners. We replayed the trajectories of BCP in *Cramped Rm.* and *Coord. Ring* and observed that BCP occasionally became immobilized and failed to collaborate with partners (Figure 4b).

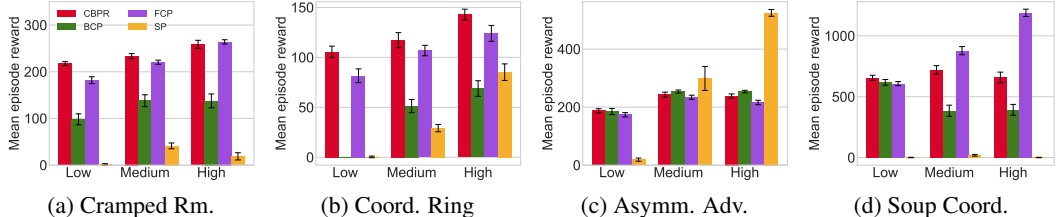

(a) Cramped Rm.   (b) Coord. Ring   (c) Asymm. Adv.   (d) Soup Coord.

Figure 4: Comparative performance analysis against baselines in cooperation with partners of diverse skill levels (low, medium and high). All agents were evaluated over 50 episodes and errors bars denote 95% confidence intervals.

## 4.3 Cooperation with Real Humans

To address question **Q3**, we recruited 25 volunteers from a local university, comprising 5 females and 20 males, ranging in age from 21 to 34 years, to participate in a study involving collaboration with CBPR and baseline agents. These volunteers were randomly assigned to one of four groups, each corresponding to a different game layout. Prior to the experiment, nearly all volunteers were unfamiliar with *Overcooked*. We provided comprehensive instructions from scratch and allowed them to play at least five practice rounds before beginning the evaluation. Subsequently, participants were instructed to interact with both the CBPR and baseline agents through the human-AI web applications developed by Carroll et al. [2019]. Each volunteer participated in two episodes, during which we recorded the average reward obtained.

According to the reward distribution (Figure 5), we observe that CBPR achieves more efficient collaboration than other baselines. In most comparisons, CBPR displays significant higher reward according to the one-sided Mann-Whitney U test.

**Case study** To further demonstrate how the CBPR is more superior than baseline algorithms when collaborating with real humans, we present a case in Figure 6. In this case, we record five frames from the *Overcooked* game interface to show that the ability of CBPR to adaptively adjust cooperative policies. Initially, CBPR agent is ready to use a dish to serve the soon-to-be-ready soup. When the human partner picks the soup, CBPR will set down the dish and continue to place onions to the pot for a new round. Meanwhile, FCP, after putting down the dish, will appear confused until the human served the soup. BCP, on the other hand, will not put down the dish and stubbornly prepare to serve the soup, ignoring the fact that the soup had already been served.

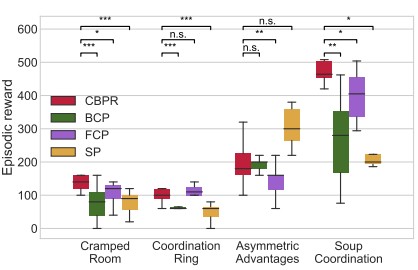

Figure 5: Rewards distribution of agents collaborating with real humans over four layouts. *, $p < 0.05$; **, $p < 0.01$; ***, $p < 0.001$, and n.s., not significant. (Statistical significance was assessed by a one-sided Mann-Whitney U test.)

## 4.4 Ablation Study

**Ablation on the queue size $l$ and inter-episodic belief weight $\rho$.** In CBPR, the length $l$ of human behavior queue and weight $\rho$ of inter-episodic belief mainly influence the collaborative performance. The larger $l$ in $P(\mathcal{Q} \mid \tau)$ of Eq. 3 means that CBPR chooses policy considering more past human behaviors. The larger $\rho$ determines that CBPR needs to consider inter-episodic belief more at the beginning of an episode. To answer question **Q4**, we expand on the experiments from section 4.2 demonstrate the results in Figure 7 and Appendix D.2. Overall, the results show that $l$=20 performs best, and in a relative simple layout (i.e., *Cramped Rm.*), since the belief of cooperative policy converges easily, variation in $\rho$ has little impact on the reward. However, in complex layout (e.g., *Soup Coord.*) (Figure 16), adjusting $\rho$ can enhance cooperative performance to a certain extent.

**Ablation on the number of predefined meta-tasks.** The performance of CBPR depends on the design of the meta-tasks. To address the challenge of predefined meta-tasks not covering all

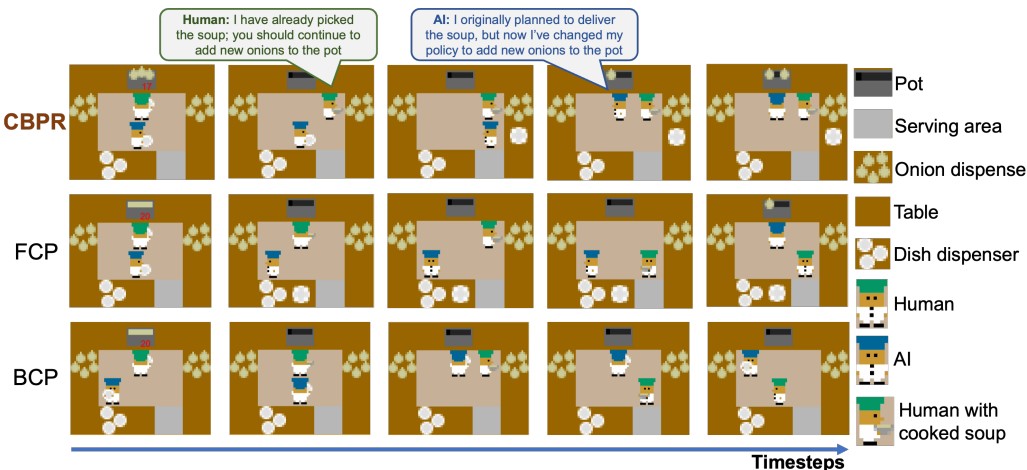

Figure 6: This case study analyzes five discontinuous frames from the *Overcooked* game interface to demonstrate the superiority of the CBPR algorithm. When a human player picks the cooked soup from the pot, the CBPR agent adapts by altering its initial plan to deliver the soup: it sets down the dish and places new onions in the pot, thereby showcasing its ability to adjust to human policies. In contrast, the FCP agent displays confusion when the human retrieves the soup and resumes placing onions only after the soup is served. The BCP agent rigidly adheres to its predetermined plan, continuously holding the plate without switching tasks to place onions, ignoring the fact that the soup has already been served.

possible ones in complex task scenarios, we introduce a meta-task category as "other" (Figure 1, bottom-right) which is represented using a random agent in practice. To demonstrate the impact of the number of predefined meta-tasks in the *Soup Coord.* We pair CBPR of different numbers of predefined meta-tasks with agents employing various skill levels. The results in Table 1 show that without "others" category, the performance deteriorates significantly, while the performances degrade relatively gracefully with less meta-tasks defined and more included in "others" category.

Table 1: Collaboration performance of CBPR with different numbers of meta-tasks and agents employing various skill levels. We report the mean reward over 10 episodes and the values in bracket represent the standard deviation. Here, we additionally define four meta-tasks (i.e., *place onion & deliver soup*, *place tomato & deliver soup*, *pickup tomato & place mix* and *pickup ingredient & place mix*), which are not included in Table 4.

|  | 7 predefined w/ "others" | 5 predefined w/ "others" | 3 predefined w/ "others" | 3 predefined w/o "others" |
|---|---|---|---|---|
| High | 620.3 (193.3) | 600.7 (234.0) | 647.7 (159.3) | 622.8 (205.8) |
| Medium | 757.8 (100.3) | 735.8 (98.7) | 717.1 (148.1) | 607.3 (278.5) |
| Low | 689.8 (43.9) | 680.5 (51.6) | 668.9 (49.0) | 40.0 (59.1) |

## 4.5 Additional Findings and Analysis

**The inherent advantage of SP and FCP agents.** Checkpoints, which are essentially SP agents, represent partners with low, medium, and high skill levels at the beginning, middle, and end of FCP training. Therefore, SP and FCP agents have an inherent advantage in the evaluation presented in Figure 4. Despite this, CBPR performs better when dealing with partners of lower skill levels. When collaborating with real humans, FCP and SP no longer hold the same advantages. This leads to almost all FCP and some SP performing well against agents of various skill levels, but falling short when facing human players.

**The cooperative advantage of CBPR in non-separated layouts.** In separated layouts (i.e., *Asymm. Adv.* and *Soup Coord.*), agents can usually complete tasks independently without considering the hindrance of the other partner's moves to themselves. However, players' own position (e.g., stand still

in front of the serving areas) can obstruct their partners from completing the task in the non-separated layouts. Therefore, non-separated layouts require more cooperation between players compared to separated layouts. As shown in Figure 4, CBPR's better performance in *Cramped Rm.* and *Coord. Ring* suggests its advantage in collaborative tasks.

**The double-edged sword of SP's simple policy.** In *Asymm. Adv.*, SP agent exhibits outstanding performance when it cooperates with the agent of high skill level (Figure 4c). We replayed the game and found that the SP agent learned the simplest and most effective policy (i.e., in the right room, just pick an onion from onion dispenser and then place it in a pot within the shortest path). On the contrary, other agents exhibit some superfluous actions due to their own complexity. However, when cooperating with the agent of low skill level, SP performs poorly because the SP agent on the right only learned the simplest policy (putting onions in the pot), and when the agent with low skill level on the left does not deliver the cooked soup, SP

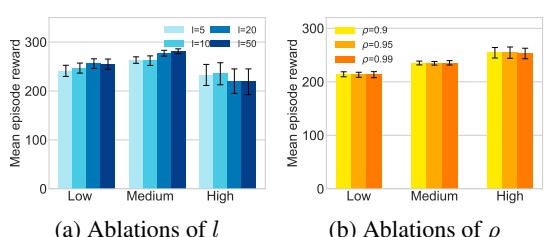

(a) Ablations of $l$      (b) Ablations of $\rho$

Figure 7: Episodic reward by using different length $l$ of human behavior queue and weight $\rho$ of inter-episodic belief in *Cramped Rm.* layout. All agents are evaluated over 50 episodes and error bars denote 95% confidence intervals.

will wait in place rather than deliver the cooked soup. In a more complex layout *Soup Coord.*, we found that the SP agent learned a policy of putting only one onion in the pot and starting to cook, leaving its partner confused and uncertain about what went wrong. Therefore, cooperation with SP agents leads to low performance (Figure 4d).

## 5  Conclusion and Discussion

**Conclusion**  In this work, we proposed CBPR framework and evaluated it in the well-known game *Overcooked*. CBPR could effectively tackle the challenge of collaborating with humans by utilizing a suite of meta-task aware agents. In response to the non-stationary nature of human behavior, CBPR adeptly selects MTP agent based on the most recent human actions and episodic returns. We have theoretically underpinned the collaborative efficacy of the CBPR approach. Empirically, we demonstrated that CBPR outperforms baselines when collaborates with simulated humans that change their policies frequently, simulated humans that employ different skill levels and real human players. We remark our primary argument that, given the non-stationary inherent in human behaviors, it is more effective to design various agents tailored to corresponding humans in different mental and behavioral states, rather than relying on a seemingly omnipotent single agent. After all, two heads are better than one.

**Limitations and future work**  In this work, meta-tasks are modeled by manually-designed rule-based policies. In real-world application domains such as assessing power system transient stability in power grid dispatching and autonomous driving, it is time consuming to design various rule-based policies. CBPR offers a viable strategy to model meta-tasks, facilitating the training of multiple specialized experts to handle distinct meta-tasks. A notable challenge, however, is the manual summarization of domain experts' meta-tasks. As a direction for future research, we are keen to address the task of clustering policies automatically based on human trajectories. While this study Zhang et al. [2023] has made strides in this direction, the clustering approach adopted therein tends to obscure semantic understanding, presenting hurdles for AI in comprehending human behaviors. Splitting human trajectories according to the key state may be a possible solution. Additionally, perceiving the acquisition of a specific class of shaped rewards by an agent as the execution of a meta-task merits future consideration. This approach also does not depend on human data or models and offers enhanced universality and interpretability.

## Acknowledgements

We are grateful to Professor Xiaohong Guan for his kind support of this work and anonymous reviewers for their insightful comments. This work was supported by the National Key R&D Program of China (2021YFB2400800).

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

# A  Proof of collaboration performance

## A.1  Proof of collaboration convergence

**THEOREM 1** (Collaboration Convergence of CBPR Agent). *Let $H_i := \{S_i^j, \pi_{hu,i}(S_i^j), R^j\}_{j=0}^{\infty}$ be a trajectory collected from a single stationary MDP $M_i$ within the overall NS-MDP $\{M_i\}_{i=1}^{\infty}$ under the human meta-task policy $\pi_{hu,i}$. Denote $\mathcal{D} := \{(i, H_i) : i \in [1, k]\}$ as a random variable representing a set of trajectories observed prior to the most recently completed stationary MDP $M_k$. Given $\mathcal{D}$, the response policy of CBPR agent could almost sure converge when interacting with a human partner, even when the human's policy is non-stationary.*

To establish the convergence of the posterior distribution, we first note that Doob's Martingale Convergence Theorem applies to our setting. Specifically, we have the following theorem:

**THEOREM 3** (Doob's Martingale Convergence Theorem). *Let $X_n$ be a martingale (or sub-martingale or super-martingale) with respect to the sequence of sigma-algebras $\mathcal{F}_n$, such that $E[\|X_n\|] < \infty$ for all $n$. If there exists a constant $C$ such that $E[\|X_{n+1} - X_n\| | \mathcal{F}_n] \leq C$ for all $n$, then there exists a random variable $X$ such that $X_n$ converges to $X$ almost surely and in $L^1$.*

With the aforementioned theorem, we can readily establish the proof of our theorem.

*Proof.* For non-stationary MDPs, demonstrating convergence involves showing that the algorithm can adapt to changing convergence points and ultimately reach them. Therefore, we will first establish the convergence property of the Bayesian update. Specifically, it will be demonstrated that the posterior distribution converges almost surely to the true parameter value. Subsequently, we will prove that, when using Bayesian updates, CBPR algorithms always converge to a fixed response policy, provided that the human policy remains unchanged before reaching the fixed response policy.

To establish the convergence of posterior distribution, we first proof that the Doob's Martingale Convergence Theorem holds for the Bayesian updating: $\beta_k(\tau) = \frac{\mathrm{P}(\sigma_k | \tau, \pi_k)\beta_{k-1}(\tau)}{\sum_{\tau' \in \mathcal{T}} \mathrm{P}(\sigma_k | \tau', \pi_k)\beta_{k-1}(\tau')}$.

Consider $\mathcal{F}_k$ as the sequence of sigma-algebras generated by observations up to time $k$. A fundamental property of Bayesian updating is that the expected value of the posterior distribution conditioned on past data equals the current posterior distribution, expressed as $E[\beta_{k+1}(\tau)|\mathcal{F}_k] = \beta_k(\tau)$. This holds because the posterior distribution $\beta_k(\tau)$ encapsulates all relevant information up to time $k$. Thus, conditioning on $\mathcal{F}_k$ accounts for all past observations, and in the absence of new data, the expected future posterior must align with the current posterior. This relationship signifies that, given the information available up to time $k$, the expectation of the next posterior does not deviate from the current posterior, establishing $\beta_k(\tau)$ as a martingale with respect to $\mathcal{F}_k$.

Moreover, the bounded nature of $\beta_k(\tau)$ within the interval $[0, 1]$ ensures that the Bayesian update satisfies the conditions of Doob's Martingale Convergence Theorem. Since $\beta_k(\tau)$ represents a probability, it is inherently bounded, which guarantees that the expected absolute change $E[\|\beta_{k+1}(\tau) - \beta_k(\tau)\| | \mathcal{F}_k]$ remains bounded. Additionally, with $E[\beta_k(\tau)] = 1$, the integrability condition required for martingale convergence is also satisfied. This combination of boundedness and integrability provides the mathematical foundation that guarantees the convergence of the sequence $\beta_k(\tau)$.

In conclusion, the sequence of Bayesian updates $\beta_k(\tau)$ adheres to the defining properties of a martingale and satisfies the conditions of Doob's Martingale Convergence Theorem through its inherent property and boundedness. As a result, we can conclude that the belief $\beta_k(\tau)$ regarding the human meta-task will converge as $k \to \infty$:

$$\mathrm{Pr}\Big(\beta_k(\tau)\Big) \to 1 \tag{7}$$

Secondly, to prove that the calculated best response policy of AI $\pi^{\star}$ converges to a fixed value as $k \to \infty$, we consider both the structure of the Bayesian update and the decision-making process in CBPR framework.

Given $\beta_k(\tau)$ converges, we note that the uncertainty about the human behavior policy $\tau$ diminishes with an increasing number of observations. The convergence of $\beta_k(\tau)$ to a specific distribution implies that the belief about the human's policy stabilizes. In mathematical terms, as $k \to \infty$, $\beta_k(\tau) \to \beta(\tau)$ for some fixed distribution $\beta(\tau)$.

Then the stabilized response policy of AI $\pi^{\star\star}$ is given by:

$$\pi^{\star\star} = \mathrm{argmax}_{\pi\in\Pi} \int_{\bar{U}}^{U^{\max}} \sum_{\tau\in\mathcal{T}} \beta(\tau)\mathrm{P}\left(U^+|\tau,\pi\right)\mathrm{d}U^+ \tag{8}$$

Here, the decision-making is a function of both the belief $\beta(\tau)$ and the expected utility $\mathrm{P}\left(U^+|\tau,\pi\right)$ for each AI response policy $\pi$. As $\beta_k(\tau)$ converges to $\beta(\tau)$, the decision-making process becomes increasingly dependent on a stable belief about the human's policy. Thus, the variability in the choice of $\pi^\star$ diminishes, leading to a convergence of $\pi^\star$ as well.

Formally, the convergence of $\pi^\star$ can be shown by demonstrating that the integral expression defining $\pi^\star$ becomes stable as $k \to \infty$. Since $\beta(\tau)$ stabilizes, the integral's value, which depends on the belief about $\tau$, also stabilizes. Consequently, by the linearity of convergence, the policy that maximizes this expression, $\pi^\star$, will almost sure converge to a fixed policy.

Given the convergence property of $\pi^\star$, the almost sure convergence for the response policy of our CBPR agent is established.

$\square$

### A.2 Proof of collaboration optimality

**THEOREM 2** (Collaboration Optimality of CBPR Agent). *Denoting* CBPR *for CBPR algorithm, let* $\rho(\pi, m) := \mathbb{E}[\int_{\bar{U}}^{U^{\max}} \mathrm{P}\left(U^+ \mid \tau(m), \pi\right)\mathrm{d}U^+]$ *be the expected return of exploiting AI policy $\pi$ with human meta-task policy $\tau(m)$ in MDP $M_m$. Given a positive integer $k$ and a set of trajectories $\mathcal{D}$ observed prior to the MDP $M_k$, it follows that for any subsequent stationary MDP $M_{k+\delta}$, we have:*

$$\Pr\left(\rho\big(\mathrm{CBPR}(\mathcal{D}), k+\delta\big) \geq \rho(\pi_k^\star, k+\delta)\right) \to 1 \tag{9}$$

*when $k \to \infty$, where $\pi_k^\star$ is the optimal response policy for human meta-task policy at MDP $M_k$.*

*Proof.* Considering the CBPR algorithm within the framework of MDPs, we define the expected return $\rho(\pi, m)$ as the integral of the probability of achieving utility $U^+$ given the AI policy $\pi$ and the human meta-task policy $\tau(m)$ in MDP $M_m$.

Assuming that the human policy library and AI policy library encompass all possible human meta-task policies and their corresponding best AI response policies. Then, we need to prove that the expected return of exploiting the CBPR algorithm's policy in any subsequent stationary MDP $M_{k+\delta}$ will be greater than or equal to that of the optimal response policy $\pi_k^\star$ at $M_k$. Formally, we can express this and derive it as follows:

$$\Pr\left(\rho\big(\mathrm{CBPR}(\mathcal{D}), k+\delta\big) \geq \rho(\pi_k^\star, k+\rho)\right)$$

$$=\Pr\left(\int_{\bar{U}}^{U^{\max}} \sum_{\tau\in\mathcal{T}} \beta(\tau)\mathrm{P}\left(U^+ \mid \tau, \pi_{\mathrm{CBPR}}\right)\mathrm{d}U^+ \geq \int_{\bar{U}}^{U^{\max}} \mathrm{P}\left(U^+ \mid \tau(k+\delta), \pi(k^\star)\right)\mathrm{d}U^+\right)$$

$$=\Pr\left(\int_{\bar{U}}^{U^{\max}} \sum_{\tau\in\mathcal{T}} \beta(\tau)\mathrm{P}\left(U^+ \mid \tau, \pi_{\mathrm{CBPR}}\right)\mathrm{d}U^+ - \int_{\bar{U}}^{U^{\max}} \mathrm{P}\left(U^+ \mid \tau(k+\delta), \pi(k^\star)\right)\mathrm{d}U^+ \geq 0\right)$$

$$=\Pr\left(\int_{\bar{U}}^{U^{\max}} \left[\beta(\tau(k+\delta))\mathrm{P}\left(U^+ \mid \tau(k+\delta), \pi_{\mathrm{CBPR}}\right) - \mathrm{P}\left(U^+ \mid \tau(k+\delta), \pi_k^\star\right)\right]\mathrm{d}U^+\right.$$

$$\left.+ \int_{\bar{U}}^{U^{\max}} \sum_{\tau\in\mathcal{T}-\{\tau(k+\delta)\}} \beta(\tau)\mathrm{P}\left(U^+ \mid \tau, \pi_{\mathrm{CBPR}}\right)\mathrm{d}U^+ \geq 0\right) \tag{10}$$

Where $\tau(k+\delta)$ represent the true stationary human meta-task policy at MDP $M_{k+\delta}$, $\pi(k^\star)$ is the best response of AI at MDP $M_k$, $\pi_{\mathrm{CBPR}}$ is the response policy generated by CBPR algorithm.

From theorem 1, we have $\Pr\left(\beta_k(\tau(k+\delta))\right) \to 1$, when $k \to \infty$.

Then we have:

$$\forall \tau \in \mathcal{T} - \{\tau(k+\delta)\}, \quad \beta(\tau) \to 0. \tag{11}$$

And thus the second term:

$$\int_{\bar{U}}^{U^{\max}} \sum_{\tau \in \mathcal{T} - \{\tau(k+\delta)\}} \beta(\tau) \mathrm{P}\left(U^+ \mid \tau, \pi_{\mathrm{CBPR}}\right) \mathrm{d}U^+ \to 0, \tag{12}$$

while the first term:

$$\int_{\bar{U}}^{U^{\max}} \left[\beta(\tau(k+\delta)) \mathrm{P}\left(U^+ \mid \tau(k+\delta), \pi_{\mathrm{CBPR}}\right) - \mathrm{P}\left(U^+ \mid \tau(k+\delta), \pi_k^\star\right)\right] \mathrm{d}U^+$$

converge to $\rho(\pi_{k+\delta}^\star, k+\delta) - \rho(\pi_k^\star, k+\delta)$. Since $\pi_{k+\delta}^\star$ is the best response policy at MDP $M_{k+\delta}$, the inequality $\rho(\pi_{k+\delta}^\star, k+\delta) \geq \rho(\pi_k^\star, k+\delta)$ would always hold. Consequently, we have $\mathrm{Pr}\big(\rho(\pi_{k+\delta}^\star, k+\delta) - \rho(\pi_k^\star, k+\delta) \geq 0\big) \to 1$, when $k \to \infty$. And we finally we achieve $\mathrm{Pr}\big(\rho(\mathrm{CBPR}(\mathcal{D}), k+\delta) \geq \rho(\pi_k^\star, k+\rho)\big) \to 1$, when $k \to \infty$. $\qquad\square$

Note that the above derivation holds when the human meta-task policy library and AI policy library encompass all possible human meta-task policies and their corresponding best AI response policies. In practice, this assumption is seldom met and is not necessarily required to be satisfied. However, we can still enable to optimality guarantee by augmenting both human and AI policy library with primitive policies $\Pi_p = \{\pi_1, \pi_2, \cdots, \pi_{|A|}\}$, where policy $\pi_i \in \Pi_p$ takes action $a_i \in A$ for all states Li et al. [2018].

## B   Environment settings

The Overcooked environment, as introduced in Carroll et al. [2019], presents a cooperative game where two players aim to complete as many orders as possible within a limited timeframe. In this study, we set the time constraint to 600 timesteps. The players navigate the environment to interact with various objects essential for order completion. An important aspect to note is that the current version of Overcooked requires an additional 'interact' action to initiate cooking in the pot, deviating from the version used in Carroll et al. [2019]. This change necessitates an adaptation of the previously collected human data, potentially affecting the performance of the BCP baseline. To align with this modification, we have adapted the latest version of the game to support auto-cooking when three ingredients are in a pot.

The environment's action space comprises the set $\{up, down, left, right, stay, interact\}$. The observation space is represented by a 96-dimensional vector, capturing each player's facing direction, absolute position, and relative positions to various game elements such as the partner, the nearest onion, pot, dish, serving area, etc. Our experiments utilize four distinct layouts as depicted in Figure 8. These layouts are chosen to illustrate a range of collaborative challenges and rewards associated with different cooking tasks. Detailed specifications of these layouts, including ingredients and reward schemes, can be found in our released code repository.

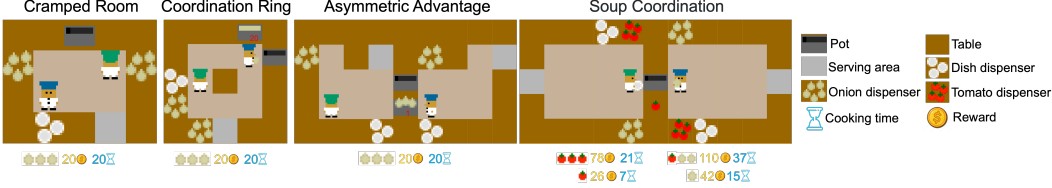

Figure 8: The four *Overcooked* experiment layouts used in our study (from left to right): *Cramped Room*, *Coordination Ring*, *Asymmetric Advantage*, and *Soup Coordination*. The game mechanics involve two players collaborating to prepare and serve dishes, like soups made of onions or tomatoes. Effective teamwork is reflected in the successful delivery of multiple orders. It is noteworthy that the *Marshmallow Experiment* layout differs from the others in terms of cooking time and reward settings.

# C  Implementation details

In our study, we rigorously implemented MTP within the CBPR framework and ensured that all baselines (BCP, FCP, and SP) adhered to a unified methodology. This approach utilized the Proximal Policy Optimization (PPO) algorithm, a widely acclaimed reinforcement learning technique Schulman et al. [2017], under a standardized set of parameters (refer to Table 2). The adoption of PPO was motivated by its balance between sample efficiency and simplicity, making it a popular choice in recent multi-agent learning research Yu et al. [2022]. To optimize the learning process and mitigate the often challenging exploration in the environment, we incorporated tailored reward shaping parameters as delineated in Table 3. This strategy aligns with the established practices in reinforcement learning that emphasize the importance of structured rewards in complex environments Gupta et al. [2022]. Additionally, our empirical analyses revealed a distinct performance advantage of feature-based observation models over the image-based ones, leading to their adoption across all agents. The entire training process was facilitated by the computational prowess of an NVIDIA 3080 GPU.

Table 2: PPO hyperparameters for MTP, BCP, FCP and SP agents. *Lambda* is used in generalized advantage estimation (GAE) to calculate advantage function. Reward shaping parameters in Table 3 gradually anneals to zero over *Reward shaping horizons*.

| Parameter | Value |
|-----------|-------|
| Learning rate | 5e-4 |
| Entropy coefficient | 0.01 |
| Epsilon | 0.05 |
| Gamma | 0.99 |
| Lambda | 0.95 |
| Batch size | 4096 |
| Clipping | 0.05 |
| Hidden dim of actor and critic | 128 |
| Reward shaping horizons | 0.5 * total timesteps |

Table 3: Reward shaping parameters for PPO.

| Action | Reward |
|--------|--------|
| Place in pot | 3 |
| Dish pickup | 3 |
| Soup pickup | 5 |

## C.1  Collaborative Bayesian Policy Reuse (CBPR)

The CBPR's offline phase is a multi-faceted process encompassing meta-task modeling, MTP, and performance modeling.

Initial efforts involved the manual definition of rule-based policies for each layout (Table 4), a step inspired by the scripted policies detailed in Yu et al. [2023].

This was followed by the training of MTP agents $\pi \in \Pi$, which were systematically paired with rule-based agents to facilitate robust policy development. The training phase, as illustrated in Figure 9, was underpinned by a commitment to capturing a diverse range of strategic interactions. Subsequently, we developed meta-task models $\tau \in \mathcal{T}$, leveraging a two-layer feed-forward neural network. This network, initialized orthogonally and optimized at a learning rate of 1e-3, was instrumental in deciphering the nuanced mappings from observations to actions.

In the final stage, performance models were crafted by pairing each MTP agent $\pi$ with rule-based meta-tasks across 50 episodes, adopting a Gaussian distribution approach to model episodic rewards.

Table 4: Predefined rule-based meta-tasks.

| Layouts | Meta-tasks |
|---|---|
| Cramped Room | 1. Place onion in pot |
|  | 2. Deliver soup |
|  | 3. Place onion and deliver soup |
|  | 4. Others |
| Coordination Ring | 1. Place onion in pot |
|  | 2. Deliver soup |
|  | 3. Place onion and deliver soup |
|  | 4. Others |
| Asymmetric Advantage | 1. Place onion in pot |
|  | 2. Deliver soup |
|  | 3. Place onion and deliver soup |
|  | 4. Others |
| Soup Coordination | 1. Place tomato in pot |
|  | 2. Deliver soup |
|  | 3. Mixed order |
|  | 4. Others |

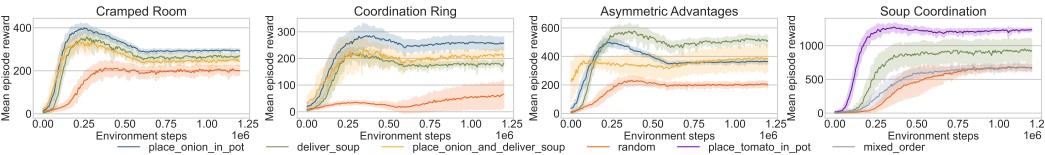

Figure 9: Training curves of meta-task playing (MTP) agents over five random seeds. The shaded area denotes the standard deviation. Noticing that the reward should not be directly compared to each other because agents vary in the partners they train with.

## C.2    Baselines

### C.2.1    Behavior Cloning (BC) and Behavioral Cloning Play (BCP) Carroll et al. [2019]

The BC models were trained using human-human trajectory data from Carroll et al. [2019]. This process, partitioning 85% of data for training and 15% for validation, aligns with the standard practices in supervised learning. The neural network, characterized by two layers with a hidden size of 64 and an orthogonal initialization, was optimized for performance with a learning rate of 1e-4 and an Adam epsilon of 1e-8. Each model underwent a rigorous 120-epoch training regimen across four layouts and five seeds, reflecting a commitment to robustness and generalizability in agent training. The BCP agents, trained in tandem with BC partners, represent a novel amalgamation of cloning and playing strategies, with training curves depicted in Figure 10.

### C.2.2    Self-Play (SP) and Fictitious Co-Play (FCP) Strouse et al. [2021]

The training of FCP agents, utilizing a pool size of 36 in the initial stage, was a strategic choice to ensure a diverse range of policy interactions. This diversity was further augmented by selecting five seeds from the first stage of FCP training for SP.

The second stage of training, involving a prolonged and intensive regimen over 50,000 episodes (amounting to 3e7 timesteps), was designed to refine and solidify the agents' strategies. Such extensive training is critical in environments characterized by high complexity and variability, as it allows agents to encounter and adapt to a wide array of scenarios. This comprehensive approach to training is evident in the detailed training curves presented in Figures 11 and 12, which provide insights into the progression and refinement of agent strategies over time.

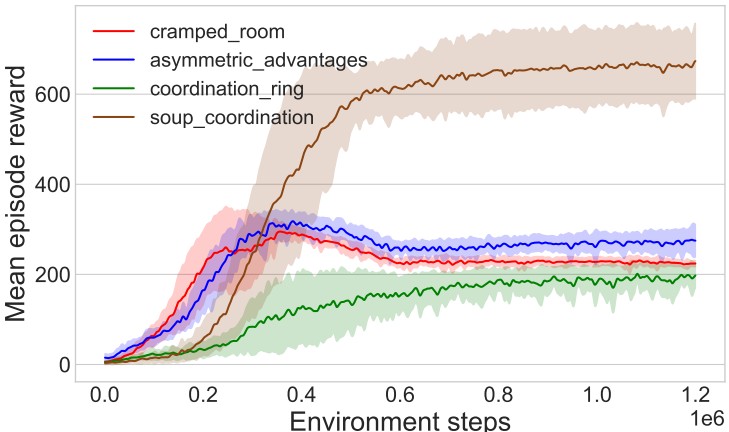

Figure 10: Training curves of BCP agents over five random seeds. The shaded area denotes the standard deviation. Noticing that the reward should not be directly compared to each other because the difficulty of the task varies with different game layouts.

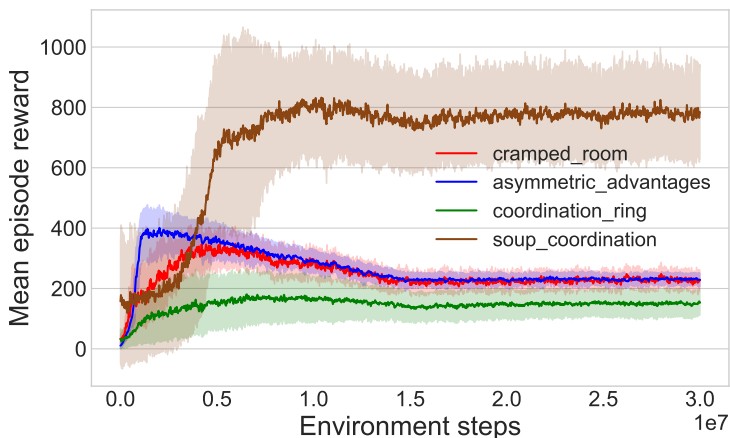

Figure 11: Training curves of FCP over five random seeds. The shaded area denotes the standard deviation. Noticing that the reward should not be directly compared to each other because the difficulty of the task varies with different game layouts.

## D  Additional results

### D.1  Collaborating with rule-based agents with various policy switching frequencies

In this section, we delve deeper into the dynamics of collaboration with rule-based agents under different policy switching frequencies. We present a series of additional experiments to complement the findings discussed in Subsection 4.1. These experiments are critical in understanding how frequent policy shifts impact the overall performance and coordination in multi-agent environments.

Figure 13 illustrates the comparative performance when rule-based agents switch policies every 2 episodes. Notably, the frequent policy changes introduce a unique set of challenges and opportunities for adaptation, as evidenced by the performance fluctuations across 50 continuous episodes. The standard error shaded areas, based on five random seeds, highlight the variability in performance under these conditions.

Similarly, Figures 14 and 15 offer insights into the performance impacts when the policy switching occurs every 200 and 100 timesteps, respectively. These results are pivotal in understanding the optimal frequency of policy shifts to achieve efficient collaboration without overwhelming the learning agents with too frequent changes.

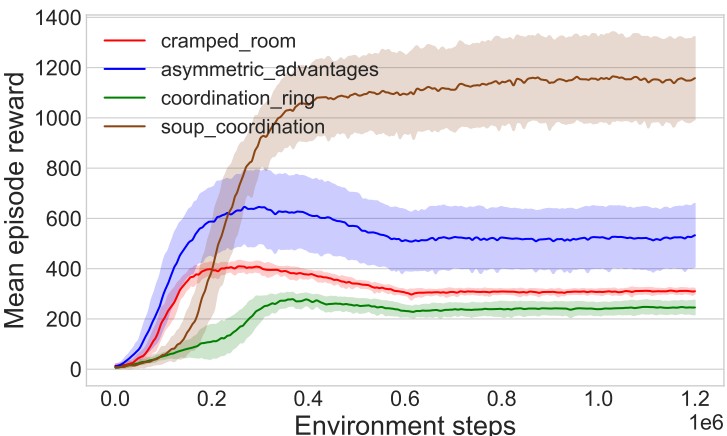

Figure 12: Training curves of self-play agents over five random seeds. The shaded area denotes the standard deviation. Noticing that the reward should not be directly compared to each other because the difficulty of the task varies with different game layouts.

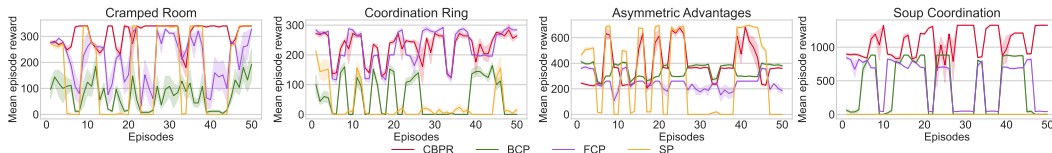

Figure 13: Comparative performance analysis against baselines when rule-based agents swith policies every *2 episodes*. All agents were evaluated over 50 continuous episodes. The shaded areas denote standard errors over five random seeds.

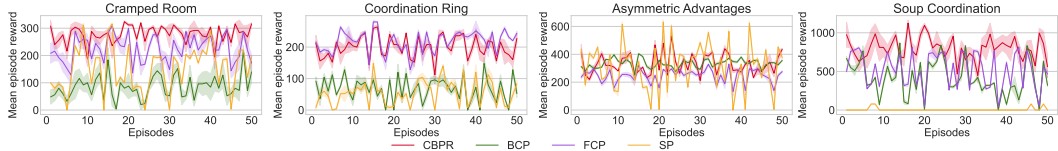

Figure 14: Comparative performance analysis against baselines when rule-based agents swith policies every *200 timesteps*. All agents were evaluated over 50 continuous episodes. The shaded areas denote standard errors over five random seeds.

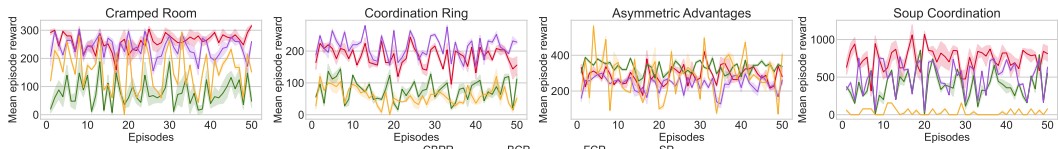

Figure 15: Comparative performance analysis against baselines when rule-based agents swith policies every *100 timesteps*. All agents were evaluated over 50 continuous episodes. The shaded areas denote standard errors over five random seeds.

## D.2 Ablation study: collaborating with partners of diverse skill levels

In the following ablation study, we focus on the aspect of collaborating with partners exhibiting diverse skill levels. This study is vital to assess how agents adapt to varying competencies within a team setting. The results of this study are shown in Figures 16 and 17, where we examine different weights and behavioral queue lengths.

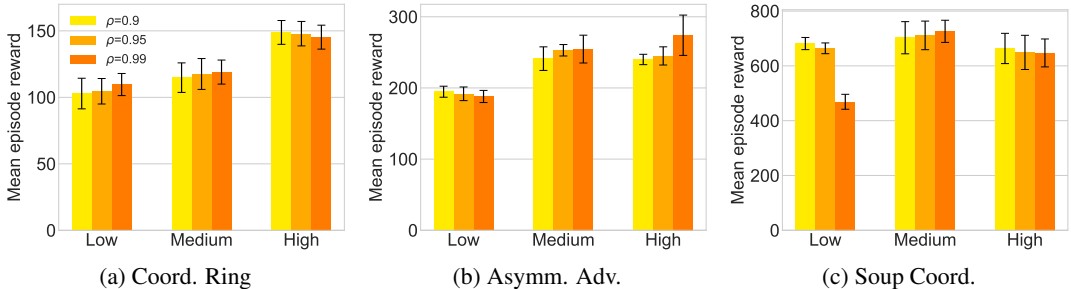

Figure 16: Episodic reward by using different weight $rho$ of inter-episodic belief in other three layouts. All agents were evaluated over 50 episodes and error bars denote 95% confidence intervals.

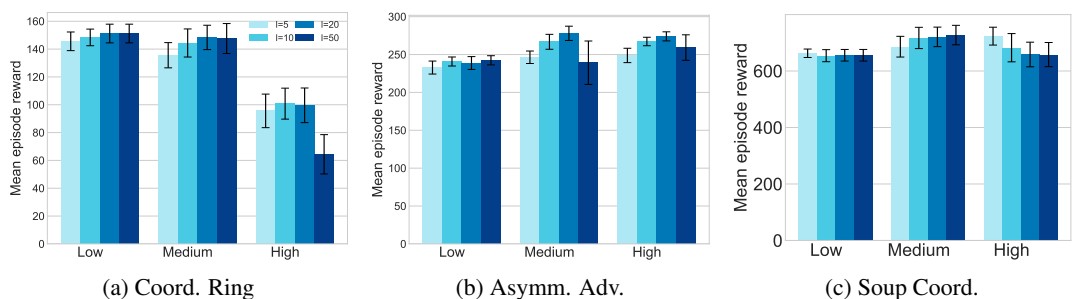

Figure 17: Episodic reward by using different length $l$ of human behavior queue in other three layouts. All agents were evaluated over 50 episodes and error bars denote 95% confidence intervals.

In Figure 16, we explore the episodic rewards obtained by varying the weight $\rho$ of the inter-episodic belief across three different layouts – *Coordination Ring*, *Asymmetric Advantage*, and *Soup Coordination*. Each layout presents a unique challenge and thus allows us to evaluate the adaptability of the agents to different team dynamics over 50 episodes. The 95% confidence intervals depicted here underscore the consistency of our findings.

Additionally, Figure 17 presents the effects of altering the length $l$ of the human behavior queue. This modification helps us understand how the memory of past interactions influences current decision-making processes in different environmental layouts. The episodic rewards over 50 episodes, along with the error bars, provide a clear depiction of the performance trends under these varied conditions.

