# OpenReview forum: "Beyond Single Stationary Policies: Meta-Task Players as Naturally Superior Collaborators"
_NeurIPS.cc/2024/Conference — NeurIPS 2024 poster_

### Official Review · Reviewer_QDdz · 2024-06-25

**Soundness:** 4
**Presentation:** 4
**Contribution:** 3
**Rating:** 8
**Confidence:** 4

**Summary:**

The authors proposes a Bayesian policy reuse-based framework referred to as CBPR, which allows for a collaborative artificial agent to adaptively select optimal collaborative policies from multiple policy networks. They did so by extending intra-episode belief to collaborative scenarios and incorporating this extension to the vanilla BPR. Their novel framework bypasses the need to model human behavior, and is suitable for human-AI collaborative tasks. The authors also provide theoretical proofs that their framework converges in cooperation with non-stationary humans, and that it can also establish the optimal collaboration policy. Finally, the authors evaluated this CBPR framework with Overcooked, a collaborative cooking game. Through experiments with this game, they demonstrated that CBPR outperforms baselines, showing that it is more effective to design multiple agents rather than a single agent.

**Strengths:**

The authors' incorporation of intra-episode belief into traditional BPR to develop CBPR was smart and original. According to their experiment with Overcooked, this novel combination of existing concepts made significant advancements in the field of human-AI collaboration. The authors went above and beyond to present the technical soundness of their work by proving their framework's convergence and optimality. Additionally, the paper is well-structured and clearly written, with meticulous attention to the setup of their experiments, enhancing the overall clarity and impact of their findings.

**Weaknesses:**

I couldn't identify any area of weakness for this manuscript.

**Questions:**

In theorem 1, the statement first mentions a stationary human policy as a given. At the end, the theorem states that convergence is also guaranteed for a non-stationary policy. It's not clear why the theorem specifically starts with non-stationary policy if it could be applied to both cases. Moreover, the proof does not clarify why the convergence guarantee for a stationary policy also applies to a non-stationary policy. Additional clarification to point out this gap would be helpful for readability.

**Limitations:**

The authors pointed out that their meta-tasks are manually-designed, rule-based policies, meaning that this method would not scale in real-world applications such as power grid dispatching and autonomous driving. While CBPR offers a strategy to model meta-tasks for human-AI collaboration, the challenge of summarizing domain experts' meta-tasks still remains.

---

> ### Author Rebuttal · Authors · 2024-08-07
>
> We are grateful for the time and effort that reviewer QDdz has invested in reviewing our paper, and we appreciate the recognition of the main advantages of our method. Additionally, thank you for your insightful comments regarding Theorem 1 and its proof.
>
> **Clarification on the Relationship Between Stationary and Non-Stationary Policies**
>
> Thank you for your thoughtful and perceptive question. Your question highlights an important aspect of our theorem formulation, and we acknowledge the need for clearer explanation in our original manuscript. Our approach is developed on the key insight that, despite the inherent
> diversity in human behaviors, the underlying meta-tasks within specific collaborative
> contexts are quite similar. This observation facilitates the transformation of a non-stationary cooperative problem into one involving a series of *stationary* policies addressing distinct meta-tasks.
>
> Theorem 1 outlines this transformation by situating the collaborative process between humans and AI within a Non-Stationary MDP framework. To manage the inherent non-stationarity, a methodical decomposition of the broader non-stationary decision-making process into *a series of temporally contiguous stationary segments, which represent the stationary process of meta-task accomplishing*.
>
> Initially, Theorem 1 defines trajectories collected from interactions with what we referred to as a 'stationary human policy.' To clarify, this term was intended to denote a stationary policy that humans use to accomplish a specific meta-task. It was not meant to imply that our theorem begins by establishing convergence for a stationary human policy and then extends this to a non-stationary human policy. We acknowledge that the term 'stationary human policy' may have led to some confusion. Thank you for your insightful query. In response, we have revised the terminology to 'stationary meta-task performing policy' in our manuscript to improve clarity and readability.
>
>
> **Clarification of Convergence in Stationary and Non-Stationary Contexts**
>
> Your question regarding convergence is astute. We believe our CBPR framework is innovative, offering a straightforward approach to achieving convergence in non-stationary human-AI cooperative settings.
>
> To ensure convergence while interacting with a non-stationary human, a cooperative policy must: 1) recognize changes in human behavior policies, and 2) adapt a consistent response policy that aligns with these changes.
>
> In the proof of Theorem 1, we first establish the convergence properties of the Bayesian update mechanism. This ensures that, with the convergence provided by Bayesian updates, the belief on the current meta-task will consistently converge. This convergence is crucial for the CBPR framework to effectively track changes in human behavior. Subsequently, we demonstrate that when utilizing Bayesian updates, CBPR algorithms consistently converge to a specific meta-task cooperative policy, provided that the meta-task being performed by the human remains unchanged until this fixed response policy is achieved.
>
> A key advantage of our CBPR method is the use of Bayesian updates to dynamically update beliefs on human behavior. This approach is effective for both stationary and non-stationary policies. The belief about the meta-task will consistently converge to match the behavior that the human is exhibiting. The primary distinction between stationary and non-stationary human policies is that the meta-task in the former remains constant, while in the latter, it evolves over time.
>
>
> We hope these clarifications address your concerns and enhance the readability and comprehension of our work. We thank you again for your valuable input, which has significantly contributed to improving our paper.

---

### Official Review · Reviewer_vD8B · 2024-07-09

**Soundness:** 3
**Presentation:** 2
**Contribution:** 3
**Rating:** 5
**Confidence:** 4

**Summary:**

This work explores how to address the challenges of non-stationary human behavior in human-AI collaboration. The authors propose a Bayesian framework that adaptively selects optimal models during training episodes to capture the underlying consistent human behavior in solving meta-tasks. Theoretical analysis shows that their proposed method can converge to optimal solutions even when the human‘s policy is non-stationary. Both simulations and real human-subject evaluations show  that the proposed method outperforms other baselines in a cooperative game Overcooked.

**Strengths:**

The motivation is realistic, and the introduction section is very clear. The proposed solution is solid, with both theoretical analysis and empirical evaluations. Sufficient details are provided for reproducibility. The idea to include both intra-episode belief and inter-episode belief is innovative.

**Weaknesses:**

The problem formulation and notations are unclear (see more discussion in Questions 1 and 2). Empirical results seem to suggest that the proposed method only makes minor improvements, and ablation studies imply that $\rho$ has almost no impact (which I believe is one of the innovations of this work). The authors do not thoroughly discuss these results. Instead, they directly conclude that the proposed method is effective.

**Questions:**

1. In line 113, $\sigma$ is defined as “any signal aiding cooperation, such as reward or interaction trajectory”. But since reward and trajectory could be derived from the given task $\tau$ and policy $\pi$, why is there uncertainty? Is the problem setup a stochastic, or is the policy stochastic?
2. In Equation (1), the belief is updated in Bayes’ rule, so the task with higher expected utility (if $\sigma$ represents reward) given current policy will be assigned a higher belief probability. Is this under the assumption that the game is cooperative and both human and AI decision-makers are acting to maximize expected utility? If not, why is such belief updating reasonable?
3. The intra-episode belief should be more stable and accurate than inner-episode belief, and contain different messages about human behavior. But from ablation studies in Section 4.2, $\rho$ seems make no difference in mean reward, why is this the case?
4. In simulation results (Figure 4), SP outperforms CBPR in asymmetric advantages, and FCP outperforms CBPR in soup coordination. Could the author discuss more about this? Similar results appear in the human experiment (Figure 6) as well. Detailed discussions about results should be added to the paper.
5. In the meta-task of Overcooked, if the player is holding an onion, then their only task is to put it in a pot; if holding a pot, the only task is to deliver it. Why is it necessary to identify the task?

**Limitations:**

No further limitations.

---

> ### Author Rebuttal · Authors · 2024-08-07
>
> We really thank you for your valuable comments on improving our work. We sincerely hope the reviewer can raise the assessment score of our paper if the following responses have successfully addressed the concerns.
>
> > Q1: The uncertainty of σ?
>
> The human-AI collaboration problem we address is inherently non-stationary. We utilized a rule-based agent with
> noise (i.e. τ ) a stochastic policy (π), thus the reward is uncertain. Signal model P (σ | τ, π), which is also known
> as observation model represents the distribution of observed signal σ when AI performs the meta-task τ with policy
> π in vanilla BPR. Note that in CBPR, we choose the episodic reward as the signal σ of signal model. During offline
> stage of CBPR, we construct each signal model (i.e. performance model) by fitting a Gaussian distribution given a
> stochastic AI policy π and a noisy rule-based agent τ (see l124-128 of the main text and l533-534 of the Appendix).
> To further clarify the definition of σ, we will add additional explanation of signal model in the revision.
>
> > Q2: The assumption of Equation 1? Why the belief updating reasonable?
>
> Thanks for the question. We would like to clarify that only the AI decision makers are acting to maximize the
> expected utility, whereas humans may exhibit varying levels of initiative to cooperate in human-AI collaboration.
> The rationale of the updating the belief in Eq 1 adheres to the vanilla BPR, which reuses prior experience from the
> class of tasks to cope with unknown task variations. Note that Following works primarily focused on extending BPR
> in competitive tasks (lines 93-95 in original paper). In this work, we firstly extend BPR in human-AI collaboration
> scenario. Human-AI collaboration is essentially an incomplete information game since human players may exhibit
> varying levels of initiative to cooperate and AI players do not have complete knowledge of payoff functions of human
> players. On the contrary, AI players need to adapt to the non-stationarity of human players, and are acting to
> maximize expected utility.
>
> > Q3: ρ seems make no difference in mean reward, why is this the case?
>
> We examine the setting of rho in the evaluation of our experiments. According to Eq 4, when ρ=0.1 the weight of the inter-episode belief decays to 0.01 after just 2 time steps, while when ρ=0.9,
> the inter-episode belief nearly decays to 0.01 within 40 time steps. Therefore, for a game with 600 time steps in
> our evaluation, only the first 40 steps of integrated belief will be influenced by the inter-episode belief, leading to
> the observation that ρ make no difference in mean reward. In other words, to demonstrate the effectiveness of ρ,
> the inter-episode belief should decay to below 0.01 after a greater number of time steps in an episode. To further
> evaluate this new setting, we set ρ = 0.99 and ρ = 0.95. Therefore, we have conducted more ablation study of ρ
> (Fig 3 in rebuttal PDF), the result indicates that in simple layout (i.e. Cramped Rm.) since the partner’s policy
> is simple, variations in ρ have little impact on the reward. However, in complex scenarios (i.e., Soup Coord.),
> adjusting rho can enhance cooperative performance to a certain extent.
>
> > Q4: Results of (Figure 4) and Figure 6.
>
> We provide a more detailed analysis of our results in Fig 4 as follows and include them in our updated paper:
>
> **The inherent advantage of SP and FCP agents.** Partners with low, medium and high skill levels are
> represented by checkpoints (essentially are SP agents) at initial, the middle and the end of training of FCP. There-
> fore, SP and FCP agents have an inherent advantage in the evaluation presented in Figure 4. Despite this, CBPR
> performs better when faced with partners of a lower skill level. When collaborating with real humans (Fig 6), FCP
> and SP no longer have such advantage. This leads to almost all FCP and some SP performing well against agents
> of various skill levels (Fig 4), but falling short when facing human players (Fig 6).
>
> **The cooperative advantage of CBPR in non-separated layouts.** In separated layouts (i.e. Asymm.
> Adv. and Soup Coord.), agents can usually complete tasks independently without considering the hindrance of
> the other partner’s moves to themselves. However, players’ own position (e.g. stand still in front of the serving
> areas) can obstruct their partners from completing the task in the non-separated layouts. Therefore, non-separated
> layouts require more cooperation between players compared to separated layouts. As shown in Fig 4, CBPR’s better
> performance in Cramped Rm. and Coord. Ring suggests its advantage in collaborative tasks.
>
> **The double-edged sword of SP’s simple policy.** In Asymm. Adv., SP agent exhibits outstanding perfor-
> mance when it cooperates with the agent of high skill level (Fig 4c). We replayed the game and found that the SP
> agent learned the simplest and most effective policy (i.e. in the right room, just pick an onion from onion dispenser
> and then place it in a pot within the shortest path) during training. However, other agents exhibit some
> superfluous actions due to their own complexity. However, when SP cooperates with the agent of low skill level, it
> performed poorly because the SP agent on the right only learned the simplest policy (putting onions in the pot), and
> when the agent with low skill level on the left does not deliver the cooked soup, SP will wait in place rather than
> deliver the cooked soup. Thus, cooperating with SP agents results in low performance (Fig 4d).
>
> > Q5: Why is it necessary to identify the task?
>
> We would like to clarify the process of ”identify” meta-task in the online stage of CBPR. Identifying task is to
> calculate the probabilities to generate a queue Q of human behaviors given each meta-task τ . During the online
> collaborating, we are unable to ascertain human cooperative strategies and initiative; however, human behaviors
> reveal the meta-tasks they are currently undertaking. Therefore, it is necessary to identify the task.

---

> > ### Comment · Reviewer_vD8B · 2024-08-09
> > **Thank you for the responses**
> >
> > Thank the authors for their detailed responses. The clarifications provided regarding the meta-task (Q1 by reviewer 3rF3) and the experimental results (Q4) have addressed most of my concerns. I would be willing to raise my rating if these discussions are incorporated into the revised manuscript.

---

> > > ### Author Response · Authors · 2024-08-09
> > >
> > > Thank you so much for your reply. We will certainly include the additional experiments and these discussions in the updated version of our manuscript.

---

### Official Review · Reviewer_3rF3 · 2024-07-17

**Soundness:** 2
**Presentation:** 2
**Contribution:** 2
**Rating:** 6
**Confidence:** 3

**Summary:**

This paper introduces Collaborative Bayesian Policy Reuse (CBPR), a framework that addresses the challenge of collaborating with non-stationary human behavior by adaptively selecting optimal collaborative policies based on the current meta-task. CBPR identifies meta-tasks underlying human decision-making and trains specialized meta-task playing (MTP) agents to enhance collaboration. Evaluations in the Overcooked simulator demonstrate CBPR's superior performance compared to existing baselines.

**Strengths:**

Overall, I found his paper seems to be interesting. It introduces the CBPR framework that addresses collaborating with non-stationary humans by matching meta-tasks rather than directly modeling complex human dynamics. The presentation of the paper could be improved. For instance, I found figure 2 to be confusing, I encourage the author could simply the diagram and improve the presentation style.

The strengths are:

1. The paper provides strong theoretical proofs and extensive empirical results across various conditions that compellingly validate the effectiveness of CBPR over baseline approaches.

2. The paper presents an approach seem to be different from the main stream: it avoids modeling human behavior but instead "focusing on constructing meta-tasks that underpin human decision making"

**Weaknesses:**

1. The effectiveness of the CBPR framework relies heavily on accurately predefined meta-tasks, which might limit its application in environments where meta-tasks are not clearly defined or are too complex to categorize effectively.

2. While the authors extensively evaluate CBPR in the Overcooked environment across various conditions, the paper lacks experiments in other domains beyond gaming. Overcooked, being a relatively simplistic and controlled environment, may not fully capture the complexities and challenges of human-AI collaboration in real-world applications such as autonomous driving, robotics, or complex decision-making systems. Testing CBPR's performance in more diverse and realistic domains would strengthen the paper's claims and demonstrate its broader applicability.

**Questions:**

1. How does the CBPR framework handle environments where human behaviors and tasks are not only non-stationary but also highly unpredictable or undefined?

2. The author used 5 different random seeds, but from the experimentation sections, looks like the variance of the run aren't that much, does this mean that the learning process is more deterministic?

3. How would the performance be like, if the episode has longer horizon?

4. How would the algorithm scales with LLMs or large model?

**Limitations:**

Yes, the author discussed the limitation and societal impact.

---

> ### Author Rebuttal · Authors · 2024-08-07
>
> We thank the time and effort reviewer 3rF3 has invested in reviewing our paper. For the Weakness1, Weakness2, Q1 and Q3 you mentioned, we have added experiments to support our insight. We apologize for the confusion caused by Figure 2. We have simplified it in the rebuttal PDF to make the framework of CBPR more clear. We provide detailed explanations for the other questions and hope this can address your concerns.
>
> > Weakness1 and Q1: The difficulty of predefined meta-tasks and the unpredictability of human behavior.
>
> For many tasks, defining meta-tasks based on human knowledge is not so difficult. For instance, in the context of autonomous driving, proactive lane changing and forward driving can serve as meta-tasks. In robotic scenarios, navigation and robotic arm grasping can also be considered as meta-tasks. To address leftover tasks not included in predefined meta tasks, we introduce a meta-task category as ”others” (Fig 1, bottom-Left).
>
> For the undefined and unpredictable tasks, they can, to some extent, be addressed by the “others” category by learning a general model for them in our current framework. To demonstrate this, we add an ablation study based on section 4.2 to examine the impact of the number of meta-tasks undefined for the Soup Coord. scenario, in which we defined 4 more meta-task (\textit{place_onion_and_deliver_soup}, \textit{place_tomato_and_deliver_soup}, \textit{pickup tomato and place mix}, \textit{pickup_ingredient_and_place_mix}). The results show that without "others" category, the performance deteriorates significantly, while the performances degrade relatively gracefully with less meta-tasks defined and more included in "others" category.
>
> | | 7 predefined+"others" | 5 predefined+"others" | 3 predefined+"others" (original paper) | 3 predefined w/o "others" |
> | :---- | :----: | :----: | :----: | :----: |
> | High | 620.3(193.3) | 600.7 (234.0) | 647.7(159.3) | 622.8(205.8) |
> | Medium | 757.8(100.3) | 735.8(98.7) | 717.1(148.1) | 607.3(278.5) |
> | Low | 689.8(43.9) | 680.52(51.6) | 668.9(49.0) | 40.0 (59.1) |
>
> In some extremely complex tasks, it may be difficult to define what a meta-task is, which will be the direction of our future research. We will explore approaches to decompose tasks into finer-grained pieces so that definitive meta-tasks can appear and become definable. We will add these discussions to limitations and future work in our revised manuscript.
>
>
>  > Weakness 2: Lacking experiments in other in real-world applications.
>
> Following your recommendation, we conducted additional experiments using the CARLA autonomous driving simulator. To acquire driving behavior data, we employed human driving models from the SUMO traffic simulator, selecting random origins and destinations for each vehicle to collect driving data. This data was used to train a Behavioral Cloning (BC) agent (Agent 1) to autonomously control the vehicle. For the Control-Based Priority Routing (CBPR) agent, we designed two meta-tasks: the "line up" meta-task, developed by selecting vehicle queuing data and training a second BC agent (Agent 2), and the "others" meta-task, utilizing BC Agent 1.
>
> We evaluated both BC and CBPR agents in a 'Queue for a Left Turn' scenario, employing a rule-based simulated human. This simulated human intervention ensured the vehicle remained in line whenever the AI attempted to change lanes. Initially, as vehicles began to queue, the AI chose to turn right and overtake. At this point, human would redirect the vehicle to maintain its position in the queue through a left turn. After a single instance of human intervention, the CBPR agent switched to the "line up" meta-task and continued to wait in the queue, whereas the BC agent repeatedly attempted to overtake by turning right, necessitating continuous human intervention. Detailed results are shown in Figure 4 of the rebuttal material.
>
> > Q2:  The learning process is more deterministic?
>
> In Appendix D, we provide the training curve of all agents (Fig 9-12) over five random seeds, which indicate the learning process is not that deterministic. The shaded area in Fig 3 is indicated by standard error, which is the standard deviation divided by the square root of the the number of seeds to represent the randomness of multiple experiments. To make the Fig 3 easier to understand, we scaled down the errors in original paper. We have replotted the Fig 3 using the standard deviation as shown in Fig 1 of rebuttal PDF file.
>
> > Q3: CBPR performance in the episode with longer horizon?
>
> Thanks for this interesting question. Following your suggestion, we increased the original horizon setting from 600 to 3000, and then had the CBPR agents cooperate with the policy-switching agents, recording the cooperative rewards under five different frequencies of policy switching. The experimental results are shown in Fig 2 of rebuttal PDF, which implies that CBPR remains highly efficient in long-horizon tasks.
>
> > Q4: CBPR scales with LLMs?
>
> Thank you for your question, which has inspired us to undertake some work combining large models with CBPR in the future. We consider CBPR scaling with LLMs or large models in following two aspects:
> \textbf{Meta-task identification} \ In \textit{Overcooked} simulator, meta-tasks are easy to defined. However, in environments where meta-tasks are not clearly defined or are too complex to categorize, the context comprehension ability of large models can deconstruct multiple meta-tasks based on behavior data demonstrated by humans.
> \textbf{Acceleration of belief $\beta_k(\tau)$ convergence} \ In an environment where two meta-tasks do not have clear boundaries. LLMs or large models can accelerate the convergence of $\beta_k(\tau)$ by adding an additional factor $\lambda$ to the numerator of Eq 1 or Eq 3. $\lambda$ can be determined by self-reflection [1] of LLM.
>
> [1]Shinn et al. Reflexion: Language Agents with Verbal Reinforcement Learning. 2023

---

> > ### Author Response · Authors · 2024-08-10
> > **Clarification of errors in the rebuttal above**
> >
> > We would like to clarify two errors in the above rebuttal:
> > 1. Due to the page limit of the rebuttal PDF file, we removed the simplified version of Figure 2, which is inconsistent with the description in above text;
> > 2. Meta-task category "others" is shown in Fig 1 bottom-Right of original paper, rather than the bottom-Left as stated in above text.
> >
> > We sincerely apologize for these errors and hope they have not caused any unnecessary confusion or affected your assessment of our paper.

---

> > > ### Comment · Reviewer_3rF3 · 2024-08-14
> > >
> > > Thanks the author for the detailed response, I have raised 1 point.

---

> > > > ### Author Response · Authors · 2024-08-14
> > > >
> > > > Thanks for your reply. We will definitly include above revisions in our updated manuscript.

---

### Author Rebuttal · Authors · 2024-08-07

We thank the time and effort reviewers have invested in reviewing our work. We have provided detailed explanations and clarifications to address your concerns regarding problem definition, experiments and discussions.

During this stage, we have supplemented the experiments to address reviewers' concerns regarding (1) a clarification of the meta-task. (2) CBPR's application in complex scenarios such as autonomous driving. (3) the performance of CBPR in long-horizon tasks. (4) the impact of inter-episode beliefs on the experimental results. Specific details can be found in the responses below the uploaded PDF file.

For other questions, we have provided point-by-point responses and have updated our paper accordingly.

We believe that incorporating your feedbacks has greatly strengthened our paper, and we hope you will agree with our improvements. We express our sincere gratitude to all the reviewers for their valuable comments on our manuscript again.

---

### Decision · Program_Chairs · 2024-09-25

**Decision:**

Accept (poster)

**Comment:**

The reviewers agreed that the paper addresses an important problem of effective human-AI collaboration when humans exhibit non-stationary dynamics, the proposed framework is novel, and the results would be of interest to the community. However, the reviewers also raised several concerns and questions in their initial reviews. We want to thank the authors for their responses and active engagement during the discussion phase. The reviewers appreciated the responses, which helped in answering their key questions. The reviewers have an overall positive assessment of the paper, and there is a consensus for acceptance. The reviewers have provided detailed feedback, and we strongly encourage the authors to incorporate this feedback when preparing the final version of the paper.